# Investigation on Adsorption of Polar Molecules in Vegetable Insulating Oil by Functional Fossil Graphene

**DOI:** 10.3390/ma16093434

**Published:** 2023-04-28

**Authors:** Suning Liang, Zhi Yang, Xianjun Shao, Yiming Zheng, Qiang Wang, Zhengyong Huang

**Affiliations:** 1State Grid Zhejiang Electric Power Research Institute, Hangzhou 310014, China; 2State Key Laboratory of Power Transmission Equipment & System Security and New Technology, School of Electrical Engineering, Chongqing University, Chongqing 400044, China

**Keywords:** graphene, functionalization, adsorption, vegetable insulating oil, polar molecule

## Abstract

As a new engineering dielectric, vegetable insulating oil is widely used in electrical equipment. Small polar molecules such as alcohol and acid will be produced during the oil-immersed electrical equipment operation, which seriously affects the safety of equipment. The polar molecule can be removed by using functional fossil graphene materials. However, the structural design and group modification of graphene materials lack a theoretical basis. Therefore, in this paper, molecular dynamics (MD) and quantum mechanics theory (Dmol^3^) were utilized to study the adsorption kinetics and mechanism of graphene (GE), porous graphene (PGE), porous hydroxy graphene (HPGE), and porous graphene modified by hydroxyl and carboxyl groups (COOH-HPGE) on polar small molecules in vegetable oil. The results show that graphene-based materials can effectively adsorb polar small molecules in vegetable oil, and that the modification of graphene materials with carboxyl and hydroxyl groups improves their adsorption ability for polar small molecules, which is attributed to the conversion of physical adsorption to chemical adsorption by the modification of oxygen-containing groups. This study provides a theoretical basis for the design and preparation of graphene materials with high adsorption properties.

## 1. Introduction

As a liquid dielectric, insulating oil plays an essential role in cooling and insulation, and is the key material used to ensure the safe and stable operation of a transformer [1]. Mineral insulating oil is a widely used liquid insulating medium in transformers, but its low flash point and poor biodegradability cannot meet the requirements of high fire protection and green environmental protection [2]. Therefore, researchers are focused on the study of a new insulating oil with a high ignition point and no pollution to replace mineral insulating oil. As a liquid insulating medium with a high flash point, high biodegradability, and regeneration, vegetable oil is regarded as an ideal substitute for mineral insulating oil [3]. However, the long-term operation of transformers can cause the aging of vegetable oils in electric, thermal, and magnetic fields, resulting in the formation of polar small molecules such as alcohols and acids. These small molecules significantly reduce the insulation performance of vegetable oil, and endanger the operation safety of transformer [4]. Therefore, it is of great significance to find a method to remove the aging products of vegetable oil in order to improve the service life of a transformer. In recent years, researchers have tried to remove polar molecules in vegetable oil by distillation, ultrafiltration, adsorption, and other methods [5,6,7]. Among them, adsorption with porous functional materials is a simple, fast, and effective way.

As a material with a large specific surface area, graphene provides more adsorption sites and can directly contact a variety of small molecules [8]. At the same time, graphene can be easily functionalized and modified. By modifying different groups, the adsorption capacity and selectivity of graphene-based materials can be effectively improved. Currently, graphene-based materials have been widely studied in the adsorption of hydrogen, impurities in water, and organic dyes [9,10,11]. This shows that graphene-based materials have great application prospects in efficient adsorption. However, there are few studies on the micro-mechanism of graphene-based materials adsorbing polar molecules in vegetable oil. With the rapid development of computer technology, researchers prefer to utilize molecular simulation technology to explore the aging products and anti-aging mechanisms of insulating oil [12]. In order to explore the adsorption of graphene materials at the molecular level, this paper analyzes the influence mechanisms of the porous structure and graphene modified with different groups on the adsorption of polar small molecules in vegetable oil with the molecular simulation method, which provides a theoretical basis for the change in macroscopic experimental phenomena.

In this paper, the molecular simulation method is employed to study the adsorption of polar molecules in functionalized modified graphene oil, mainly focusing on the diffusion characteristics of small molecules in oil and the dynamic process of diffusion to the interface of functionalized modified graphene, and the interaction characteristics of functionalized modified graphene and polar small molecules. The adsorption sites, adsorption energy, as well as the density of state changes are studied by the quantum mechanics method.

## 2. Computational Method and Physical Model

The model construction is all based on the Materials Studio (MS) software. The calculation process includes molecular dynamics and quantum mechanics in order to analyze the influence of the porous structure and different groups. A micro-computing model is constructed, which consists of water, formic acid, acetic acid, formaldehyde, acetaldehyde, and other polar molecules produced by the aging and cracking of graphene, triglyceride (the main component of vegetable oil), and insulating oil. Four configurations of graphene (GE), porous graphene (PGE), porous hydroxy graphene (HPGE), and porous graphene modified by hydroxyl and carboxyl groups (COOH-HPGE) were selected in this paper, as shown in Figure 1. The size of graphene is 30 Å × 30 Å, with a pore radius of 8 Å was selected by comparing the adsorption capacity of graphene with different pore sizes. Hydroxyl groups are modified on the surface of graphene, and carboxyl groups are modified on the edge of graphene. Triglycerides are mainly composed of linoleic triglycerides, oleic triglycerides, and palmitic triglycerides, with a molecular ratio of 6:3:1 [13]. The initial model box size is 32 Å × 32 Å × 96 Å, including 1 piece of graphene, 20 triglycerides, and 25 polar small molecules. For the initial model, the unreasonable conformation is eliminated through the processing of Geometry Optimization and Annex. Selecting the COMPASS force field [14], which is universal for graphene materials, and controlling the pressure and temperature, respectively, through NPT and NVT ensembles, allows us to obtain a stable system. Finally, the interaction energy, hydrogen bond, and other microscopic parameters are calculated and analyzed.

The Dmol^3^ module based on quantum mechanics is selected to build a small molecular model of graphene adsorption polarity. Figure 2 shows the small molecular diagram of graphene adsorption polarity. The size of the graphene sheet is 13.5 Å × 13.5 Å, and hole radius is 3.3 Å. For the adsorption model, the density functional of the optimization process is in the form of GGA-PBE, the valence electron wave function is in the form of DNP, and the effect of spin polarization is considered in all calculations. The adsorption energy, differential charge, and density of states of the optimized structure are obtained through energy analysis [9]. The calculation formula for adsorption energy is as follows:$$E_{\mathrm{ads}}=E_{\mathrm{total}}-E_{\mathrm{GE}}-E_{\mathrm{polar}}$$
where *E*_ads_ is the adsorption energy; *E*_total_ is the total energy of polar small molecules adsorbed on graphene; *E*_GE_ is the energy of graphene; *E*_polar_ is the energy of small polar molecules.

## 3. Results and Discussion

### 3.1. Diffusion Characteristic

The motion process of polar small molecules can be more easily seen from the microscopic scale. Figure 3 is a snapshot of the dynamic movement of polar small molecules in different systems (oil molecules are hidden). In the initial conformation, the polar molecules of the four systems are uniformly dispersed. However, after the dynamic calculation, the uniformly dispersed state is changed, and the polar molecules tend to move towards the surface of graphene, which indicates that graphene-based materials have a strong interaction and adsorption capacity for polar molecules in insulating oil. Meanwhile, under the action of the porous structure and modified groups, a small number of polar small molecules pass through the carbon pores, and the polar molecules gather on the surface of graphene. This means that graphene modification is beneficial to enhance the adsorption capacity of graphene-based materials to polar small molecules.

### 3.2. Interaction Energy

The interaction energy of graphene-based materials and vegetable oil, and graphene-based materials and polar small molecules in the four models were calculated (see Appendix A), respectively. It can be seen from Figure 4 that the interaction energy of all systems is negative, and the increase in interaction energy leads to stronger adsorption and more stable structure [15]. With the complexity of modification, the interaction between graphene-based materials and vegetable oil or polar small molecules is stronger. Compared with the interaction energy between graphene-based materials and vegetable oil and their polar molecule pairs, the adsorption capacity of GE or PGE to vegetable oil is higher than that of polar molecule pairs, which also conforms to the experimental results [16,17]. However, the adsorption capacity of HPGE and COOH-HPGE to polar small molecules increased significantly. E_NEIO/HPGE_ (−402.51 kcal/mol) and E_NEIO/COOH-HPGE_ (−242.97 kcal/mol) are 18.69% and 97.40% higher than E_NEIO/GE_ (−203.93 kcal/mol), and E_PSM/HPGE_ (−374.61 kcal/mol) and E_PSM/COOH-HPGE_ (−907.06 kcal/mol) are 892.08% and 2302.17% higher than E_PSM/GE_ (37.76 kcal/mol), respectively. This means that the modification of oxygen-containing groups can simultaneously increase the adsorption capacity of graphene-based materials to vegetable oil and polar small molecules, but the GE rich in oxygen-containing groups is polar, so its adsorption selectivity to polar small molecules is significantly enhanced, especially under the action of hydroxyl and carboxyl groups.

Hydrogen bonds are an important factor enhancing the interaction between substances [18]. It can be seen from Figure 5 that the number of hydrogen bonds in the graphene system without group modification is about 38. These hydrogen bonds are attributed to the hydrogen bonds between polar small molecules rather than the hydrogen bonds at the interface. The strong electrostatic attraction of highly polarized functional groups greatly increases the number of hydrogen bonds between graphene-based materials and other molecules, which makes other molecules inevitably move to the surface of graphene in motion.

### 3.3. Adsorption Characteristics

Quantum mechanics can better explain the role of oxygen-containing groups in the adsorption process. In this paper, we used the Dmol3 module based on quantum mechanics to calculate the adsorption energy of single polar small molecules in vegetable oil on graphene, and to observe their optimized structures. From the optimized structure (Figure 6), it can be seen that the acetic acid molecule is 3.033 Å and 3.256 Å away from the GE and PGE surfaces, respectively, indicating that GE and PGE have a small adsorption energy for the acetic acid molecule. In contrast, the acetic acid molecule tends to adsorb on the oxygen-containing groups of HPGE and COOH-HPGE, with adsorption distances of 1.222 Å and 1.481 Å, respectively, which means that HPGE and COOH-HPGE have enhanced adsorption capacity for the acetic acid molecule compared to GE and PGE [19]. Additionally, the interactions between the acetic acid molecule, hydroxyl group, and carboxyl group form a complex adsorption structure, which may be the reason why COOH-HPGE has a greater adsorption energy for polar small molecules. To more accurately reflect the adsorption capacity of the four graphene-based materials for polar small molecules, the adsorption energies between them were calculated. From Table 1, as the content of oxygen-containing groups increases, the adsorption energy between graphene-based materials and polar small molecules gradually increases. It is generally believed that adsorption energies less than 0.25 eV are physical adsorption, and those greater than 0.25 eV are chemical adsorption [20]. It is worth noting that the adsorption energies of GE and PGE for polar small molecules are both less than 0.25 eV, and the distance between the acetic acid molecule and GE and PGE is far greater than the bond length. In contrast, the adsorption energies of HPGE and COOH-HPGE for polar small molecules are both greater than 0.25 eV, and the distance between the acetic acid molecule and the HPGE and COOH-HPGE surfaces is relatively close. This indicates that the low adsorption of polar small molecules by GE and PGE is physical adsorption, and the higher adsorption of HPGE and COOH-HPGE for polar small molecules may be caused by chemical adsorption. To further verify the adsorption type of graphene-based materials for polar small molecules, the charge difference density of the optimized structure was analyzed. In Figure 7, the red area represents an increase in charge, and the blue area represents a decrease in charge [21,22]. In the GE and PGE systems, the acetic acid molecule only undergoes intramolecular electron transfer, while the oxygen-containing groups of HPGE and COOH-HPGE undergo obvious charge transfer with the acetic acid molecule (at the arrow). Based on the above results, the high adsorption energy, short adsorption distance, and charge transfer confirm that the adsorption of HPGE and COOH-HPGE for polar small molecules belongs to chemical adsorption.

The energy level interactions between the polar small molecules and graphene-based materials during the adsorption process can be described by comparing the density of states (DOS) before and after adsorption. Figure 8 shows the DOS of the four systems near the Fermi level (−3 eV to 3 eV). The DOS of GE and PGE before and after the adsorption of acetic acid molecules are basically overlapping, and the only difference is that two peaks in the interval of −2.0 to −0.5 merge into a broad peak, which may be caused by the enhanced electronic delocalization of the acetic acid molecule after adsorption. Compared with the GE system, PGE, HPGE, and COOH-HPGE show multiple larger bandgaps, indicating that pore structure and oxygen-containing group modification of graphene change its electronic state, which is attributed to the disturbance of the symmetry of graphene by the oxygen-containing groups. After the adsorption of acetic acid molecules, the DOS peak curve shifts towards the lower energy direction (left), which means that the adsorption structure becomes more stable. At the same time, the height of the DOS peak decreases significantly, making it easier for electrons to transition, thus promoting the charge transfer between the graphene-based substrate and the acetic acid molecule.

## 4. Conclusions

This article uses molecular dynamics (MD) and quantum mechanical theory (Dmol3) simulation methods to compare the dynamic snapshots, interaction energy, hydrogen bonds, adsorption energy, differential charge, and density of states (DOS) of four materials, namely graphene (GE), porous graphene (PGE), porous hydroxylated graphene (HPGE), and hydroxyl and carboxyl co-modified porous graphene (COOH-HPGE), for polar small molecules (PSMs) in vegetable oil (NEIO). Based on MD analysis, compared with the GE and PGE models, PSMs tend to aggregate on the surface of oxygen-containing functionalized graphene due to the increased number of hydrogen bonds. Meanwhile, HPGE and COOH-HPGE exhibit significantly enhanced interaction energy with PSMs, with EPSM/HPGE (−374.61 Kcal/mol), and EPSM/COOH-HPGE (−907.06 Kcal/mol), respectively, increasing by 892.08% and 2302.17% compared with EPSM/GE (37.76 Kcal/mol). Based on Dmol3 analysis, the adsorption energy of graphene-based materials on PSMs gradually increases with the complexity of modification. The adsorption of PSMs on GE and PGE is confirmed to be physical adsorption only, while the adsorption of PSMs on HPGE and COOH-HPGE is confirmed to be chemical adsorption, indicated by the adsorption energy, adsorption distance, and differential charge transfer. Finally, the DOS analysis further confirms that the structure of graphene-based materials modified with oxygen-containing functional groups is more stable, and electron transition is easier when adsorbing polar molecules. This study provides a theoretical analysis method for designing functionalized graphene materials with the ability to remove PSMs from vegetable oil.

## Figures and Tables

**Figure 1 materials-16-03434-f001:**
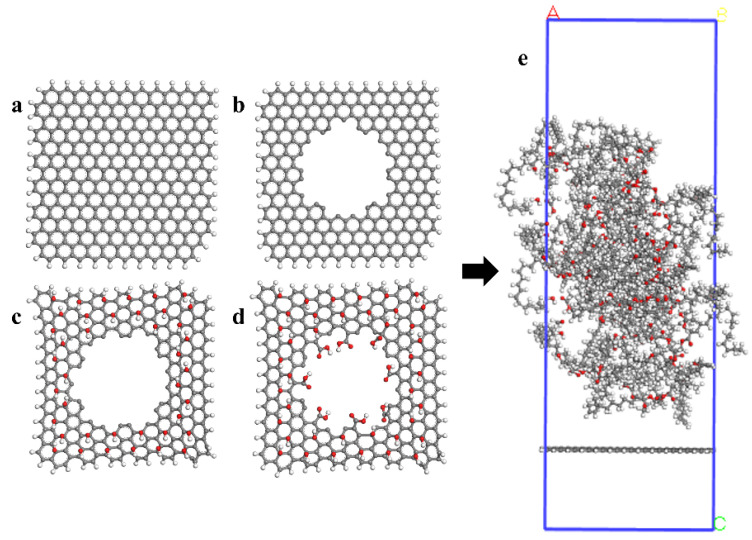
Kinetic model of graphene-based materials adsorbing polar small molecules in vegetable oil: (**a**) GE; (**b**) PGE; (**c**) HPGE; (**d**) COOH-HPGE; (**e**) snapshot of the initial model.

**Figure 2 materials-16-03434-f002:**
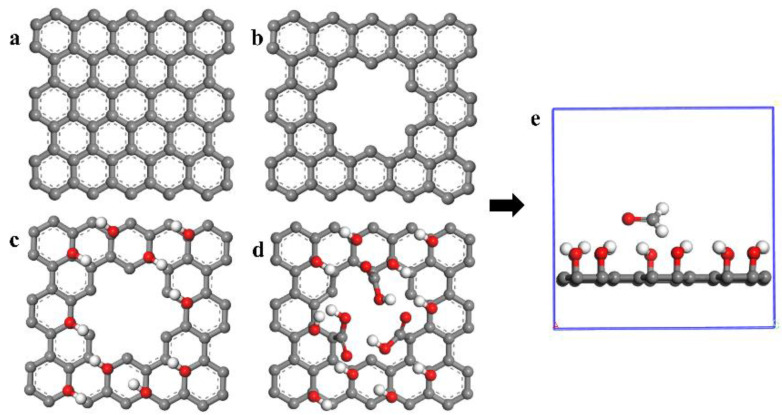
Quantum mechanical model of graphene-based materials adsorbing polar molecules: (**a**) GE; (**b**) PGE; (**c**) HPGE; (**d**) COOH-HPGE; (**e**) snapshot of the initial model.

**Figure 3 materials-16-03434-f003:**
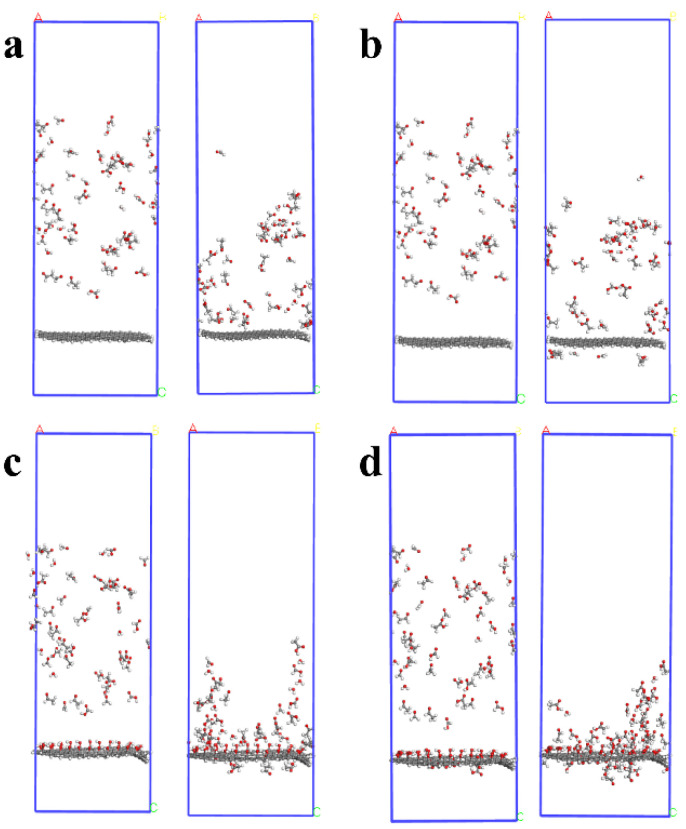
Snapshots of polar molecules before and after the dynamics of the four systems: (**a**) GE; (**b**) PGE; (**c**) HPGE; (**d**) COOH-HPGE.

**Figure 4 materials-16-03434-f004:**
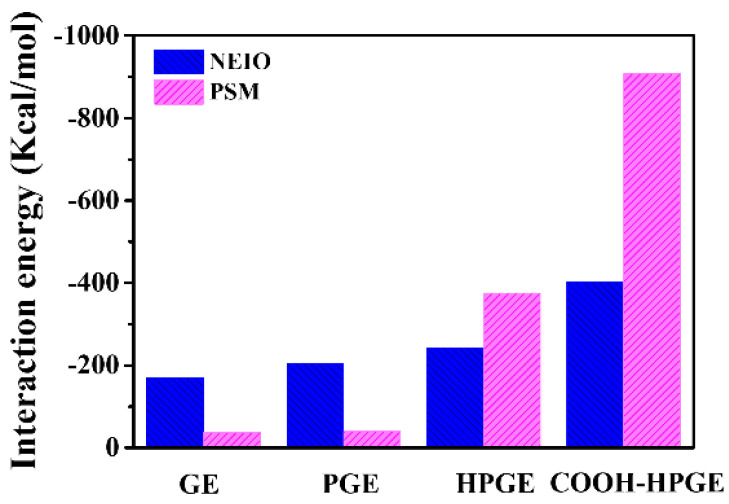
Interaction energy of different graphene systems.

**Figure 5 materials-16-03434-f005:**
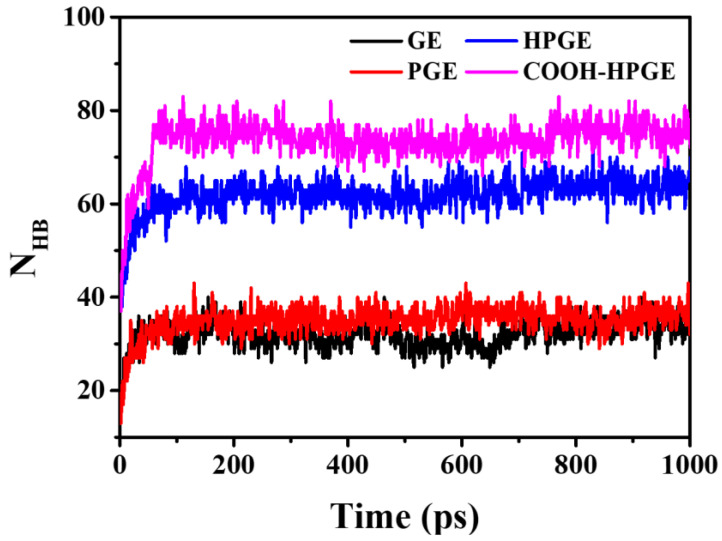
Number of hydrogen bonds of different graphene-based systems.

**Figure 6 materials-16-03434-f006:**
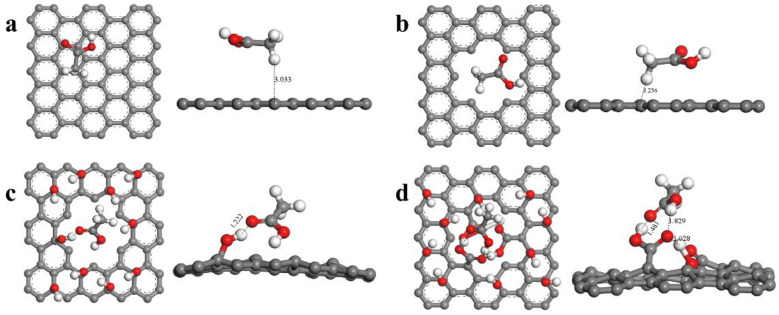
The optimized structure of adsorbing polar molecules (taking acetic acid as an example): (**a**) GE, (**b**) PGE, (**c**) HPGE, (**d**) COOH-HPGE.

**Figure 7 materials-16-03434-f007:**
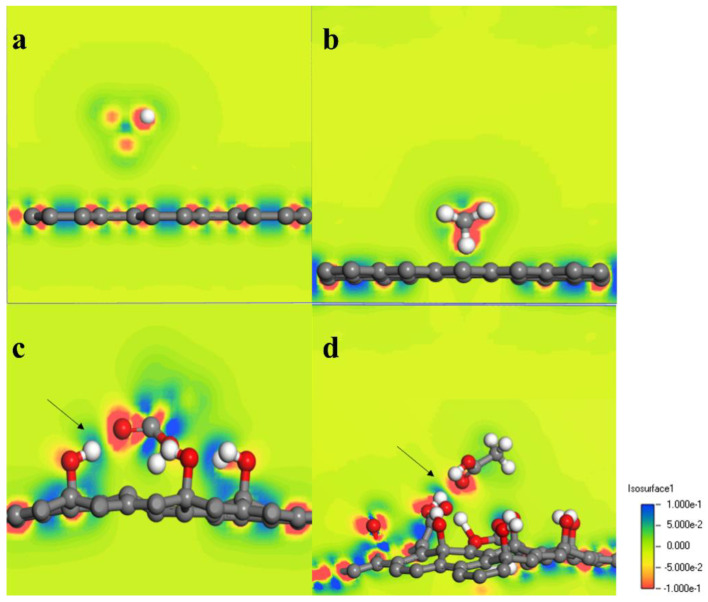
Differential charge diagram of adsorbed acetic acid small molecule: (**a**) GE, (**b**) PGE, (**c**) HPGE, (**d**) COOH−HPGE.

**Figure 8 materials-16-03434-f008:**
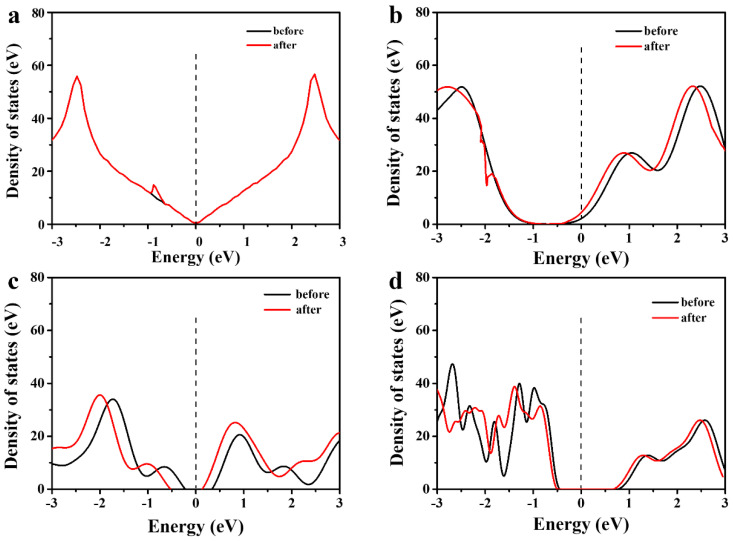
DOS diagram of adsorbed acetic acid small molecules: (**a**) GE, (**b**) PGE, (**c**) HPGE, (**d**) COOH−HPGEDOS.

**Table 1 materials-16-03434-t001:** Adsorption energy of graphene-based materials for adsorbing polar small molecules (eV).

	Graphene Based	GE	PGE	HPGE	COOH-HPGE
Small Molecules					
H_2_O		−0.121	−0.157	−1.219	−1.757
HCOOH		−0.126	−0.172	−1.030	−1.660
CH_3_CHO		−0.186	−0.235	−0.923	−1.055
HCHO		−0.128	−0.175	−1.006	−1.634
CH_3_COOH		−0.172	−0.202	−0.998	−1.557

## Data Availability

Not applicable.

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
