# Peer review of "Investigation on Adsorption of Polar Molecules in Vegetable Insulating Oil by Functional Fossil Graphene"

_materials, 2023, doi:10.3390/ma16093434_

Round 1
Reviewer 1 Report
Review report of ‘’Investigation on adsorption of polar molecules in vegetable insulating oil by functional fossil graphene’’.
The concept of paper is to use modified structure of Graphene to remove small polar molecules during the insulation of an electric equipment. Removal of polar molecules is a serious concern in terms of safety of oiled immersed electrical equipment. Removal process is observed by physical or chemical adsorption. This work is devoted to tailor functionalization of Graphene for applications in polar molecules removal. This study provides a theoretical basis for the design and preparation of graphene materials with high adsorption properties. This theoretical prospective will provide the scientific community prospects of modified graphene and graphene oxide.
This article uses molecular dynamics (MD) and quantum mechanical theory (Dmol3) simulation methods to compare the dynamic snapshots, interaction energy, hydrogen bonds, adsorption energy, differential charge, and density of states (DOS) of four materials, graphene (GE), porous graphene (PGE), porous hydroxylated graphene (HPGE), and hydroxyl and carboxyl co-modified porous graphene (COOH-HPGE), for polar small molecules (PSMs) in vegetable oil. The DOS analysis further confirms that the structure of graphene-based materials modified with oxygen-containing functional groups is more stable and electron transition is easier when adsorbing polar molecules.
Comments on some details:
1.There are many studies published in recent years for functionalized graphene used for the removal of polar organic dyes and aromatics in the solution phase. Here the adopted theoretical methodology is novel.
2. I wonder if the enhanced defect levels in graphitic skeleton are considered while performing simulations. Author may clarify this.
3. Authors have chosen Four configurations of graphene in given set of study, I wonder if this is considered monolayer or multi-layer. In case of monolayer, this study is not applicable for cost effective methods of Graphene synthesis. Also, how multilayers are going to change the mechanism of adsorption can be studied further.
Page4, para
Acronym EGE used is not clear.
This study provides a theoretical analysis method for designing functionalized graphene materials with the ability to remove polar molecules from vegetable oil. I find that this article has value in that it has enumerated a new theoretical investigation procedure that can give guidance to researchers to work primarily.
.

Reviewer 2 Report
The paper describes the results of theoretical modelling of adsorption of polar molecules on graphene-based materials. The results are obtained via a combination of MD and DFT methods. The paper is well-written and well motivated. I still have a few questions:
1) In sec 3.1 the main advantage of pores is stated as the possibility for the molecules to stick to the back side of graphene. Can the same be achieved if simply a finite-size graphene flake is considered?
2) In figure 4, it would be better to use full names in the legend instead of abbreviations.
3) There is, obviously, a different adsorption energy of polar molecules and oil to the surface of graphene and to its functionalized edges (or pore edges). This needs a clear classification. For example, if one changes the area to boundary ratio of the modelled sample, how will the total energy change? Can the energy contributions be split into the surface and the boundary ones?
4) It seems that the DOS of graphene is calculated for a rather small system, so, it contains a lot of peaks instead of a continuum. Normally, to model a flake larger than the simulated cell, one needs to impose periodic boundary conditions at the edges of the considered cell, with an additional phase twist by Exp[i kx + i ky] and then integrate over kx and ky (or, sum over a mesh in a numerical setting). This would give a continuous density of states for graphene.
5) Returning to question 3), it could be beneficial to analyse, how many PSM can be absorbed on a given surface or edge of graphene, and how this scales with the area / and edge length.
To conclude, the paper studies a hot and interesting topic, but a deeper theoretical analysis of the numerical results would be beneficial.
Reviewer 3 Report
This manuscript employ theoretical calculation methods to study interaction mechanism of vegetable oil in various material system. Overall, this manuscript is very well written, underlying physics and chemistry are well discussed. I only have one comment: Selectivity is the most concerned factor in any nonmaterial sensing system. For graphene, such a high surface to volume ratio material, any environment disturbance could be sensed by graphene. Can the author comment on the challenges of implementing such sensing platform to the detection vegetable oil from the selectivity point of view?
Round 2
Reviewer 2 Report
I am glad that my comments helped the authors to improve the manuscript.